# Probable Causes of Alzheimer's Disease

James David Adams 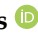

Independent Researcher, La Crescenta, CA 91214, USA; jadams@usc.edu

**Abstract:** A three-part mechanism is proposed for the induction of Alzheimer's disease: (1) decreased blood lactic acid; (2) increased blood ceramide and adipokines; (3) decreased blood folic acid. The age-related nature of these mechanisms comes from age-associated decreased muscle mass, increased visceral fat and changes in diet. This mechanism also explains why many people do not develop Alzheimer's disease. Simple changes in lifestyle and diet can prevent Alzheimer's disease. Alzheimer's disease is caused by a cascade of events that culminates in damage to the blood–brain barrier and damage to neurons. The blood–brain barrier keeps toxic molecules out of the brain and retains essential molecules in the brain. Lactic acid is a nutrient to the brain and is produced by exercise. Damage to endothelial cells and pericytes by inadequate lactic acid leads to blood–brain barrier damage and brain damage. Inadequate folate intake and oxidative stress induced by activation of transient receptor potential cation channels and endothelial nitric oxide synthase damage the blood–brain barrier. NAD depletion due to inadequate intake of nicotinamide and alterations in the kynurenine pathway damages neurons. Changes in microRNA levels may be the terminal events that cause neuronal death leading to Alzheimer's disease. A new mechanism of Alzheimer's disease induction is presented involving lactic acid, ceramide, IL-1β, tumor necrosis factor α, folate, nicotinamide, kynurenine metabolites and microRNA.

**Keywords:** lactic acid; ceramide; folate; nicotinamide; Alzheimer's disease

## 1. Introduction

The failure of anti-amyloid β therapies in the treatment of Alzheimer's disease [1–5] has pointed out that the amyloid-tau theories of Alzheimer's disease induction are wrong. As many Scientists have published for many years, amyloid β and tau are not the causes of Alzheimer's disease. Amyloid β and tau are misfolded proteins that accumulate in neurons and may be released into extracellular spaces when neurons die.

There are several risk factors for the development of Alzheimer's disease, including loss of muscle mass, high alcohol consumption, type 2 diabetes, high blood pressure, high blood cholesterol, heart disease, atrial fibrillation and other factors [6–8]. The variables that are common to all these risk factors are sedentary lifestyles and visceral fat obesity. As people age and become sedentary, visceral fat increases, muscle mass decreases and vascular damage increases. Cerebrovascular damage is prominent in the brains of Alzheimer's disease patients [9]. Alcohol consumption increases visceral fat accumulation.

Several research articles have found that Alzheimer's disease can be prevented by eating diets low in fat, high in fruits and vegetables, and by regular physical activity [10–12]. The health benefits of fruits, vegetables and exercise cannot be overemphasized in cultures where meat-eating and sedentary lifestyles are abundant.

## 2. Exercise/Lactic Acid

Decreased blood lactic acid is involved in causing Alzheimer's disease. There is a muscle brain axis that is based on the fact that the brain must have substances that are secreted into the blood by muscles. This muscle brain axis is critical to brain health. Loss of muscle mass with aging is dangerous for the brain. Sarcopenia is a normal consequence of

aging, even in athletes [13]. It is likely that physical activity decreases the risk of developing Alzheimer's disease [14].

Lactic acid comes from muscle, especially exercising muscle [13]. Lactic acid is very important to the body, including the brain (Table 1). Blood lactic acid levels can reach 32 mM in athletes. Lactic acid is an energy source for brain neurons, astrocytes and pericytes [15]. It is taken up across the blood–brain barrier and into cells by anion channels and mono-carboxylic acid transporters [16]. Due to sarcopenia and sedentary lifestyles, many people may not get enough lactic acid to supply essential energy to brain cells, especially cells of the blood–brain barrier.

Hydroxycarboxylic acid receptor1 is the receptor that mediates lactic acid signaling and is abundant in cells of the blood–brain barrier [16]. The receptor is found on both the luminal and abluminal surfaces of endothelial cells of the blood–brain barrier. Autophagy and mitophagy can be inhibited by lactic acid [17]. Alzheimer's disease appears to be associated with improperly functioning autophagy [18]. Physical exercise enhances brain levels of brain-derived neurotrophic factors, vascular endothelial growth factor and lactic acid, all of which stimulate neurogenesis, even in the adult hippocampus [17,19,20]. Lactic acid improves brain health and neurogenesis but may not stimulate memory retention [20].

Lactic acid inhibits transient receptor potential cation channels of the vanilloid type (TRPV1) [19]. Endocannabinoids are the normal agonists for TRPV1 receptors and also inhibit these receptors after prolonged stimulation [21]. Lactic acid augments the inhibitory actions of endocannabinoids at TRPV1 channels. TRPV1 channel overstimulation increases calcium influx into cells and enhances oxygen radical production [22]. There are three aspects of lactic acid that are important to the blood–brain barrier. It is an essential nutrient to pericytes, astrocytes and neurons. It can inhibit autophagy. It can also inhibit TRPV1 channels, which may decrease damage to the blood–brain barrier. Pericytes and endothelial cells of the blood–brain barrier are regulated, in part, by TRPV1 channels [23,24]. Damage to pericytes and the blood–brain barrier are prominent features of Alzheimer's disease [25]. Inadequate lactic acid causes pericytes to die. Inadequate inhibition of TRPV1 by low levels of lactic acid causes pericytes to die. Pericytes secrete pleiotrophin that is essential for neuronal survival [26].

Exercise stimulates oxygen radical production, even in the brain [27]. Oxygen radical production includes the generation of hydrogen peroxide, which is an important signaling molecule. Hydrogen peroxide crosses membranes at peroxiporin channels and rapidly damages nuclear DNA, which results in poly (ADP-ribose) polymerase activation [28]. This enzyme is responsible for increasing the activities of DNA repair enzymes and enzymes that protect against oxidative stress. The transcription of nuclear factor erythroid 2 related factors 2 increases, which activates genes of the antioxidant response elements [29]. These gene sequences enhance the transcription of several enzymes involved in defense against oxidative stress. Exercise enhances the ability of the brain to protect itself against oxygen radical toxicity and enhances brain health.

Exercising muscles secrete myokines into the blood. These proteins have a number of beneficial functions in the brain and the blood–brain barrier. Cathepsin B is a myokine that crosses the blood–brain barrier and enhances the synthesis of brain-derived neurotrophic factors in the brain [20,30]. Fibroblast growth factor 21 is a myokine that crosses the blood–brain barrier, has neuroprotective effects and enhances circadian rhythms [20]. Irisin is derived from exercising muscles, crosses the blood–brain barrier, increases synaptic plasticity and memory [31].

**Table 1.** Brain muscle axis. The effects of exercise on the brain and the blood–brain barrier.

| Factor | Receptor | Effect |
|---|---|---|
| Lactic acid | Anion channels, monocarboxylic acid transporters, hydroxycarboxylic acid receptor 1 | Essential nutrition for neurons, astrocytes, pericytes |
| Lactic acid | Transient receptor potential cation channel vanilloid1 | Inhibition of the channel and oxygen radical formation |
| Hydrogen peroxide | Poly (ADP-ribose) polymerase, antioxidant response elements | At low levels enhances brain-protective mechanisms, at high levels damages DNA and other macromolecules |
| Cathepsin B | unknown | Increases brain-derived neurotrophic factor synthesis |
| Fibroblast growth factor 21 | FGF21 receptor | Neuroprotective enhances circadian rhythms |
| Irisin | $\alpha$V integrin receptor | Enhances synaptic plasticity and memory |

## 3. Ceramide

High levels of ceramide in the blood and brain are involved in causing Alzheimer's disease. Ceramide levels are high in the brains of Alzheimer's disease patients [32]. The higher the ceramide levels, the greater the risk of exhibiting Alzheimer's disease [33–35]. Ceramide is released into the blood by visceral fat adipocytes and is taken up into the brain [36–39]. Ceramide is an alternate storage form for fatty acids instead of storage in triglycerides.

Endothelial nitric oxide synthase is induced by ceramide but is dysfunctional in the presence of high ceramide levels [36–39] with the production of oxygen radicals. This damages endothelial cells of the blood–brain barrier and may allow the penetration of monocytes/macrophages and neutrophils into the brain. Ceramide-induced oxidative stress increases NADPH oxidase activity on the outside of the plasma membranes of macrophages in the brain, which produces hydrogen peroxide and increases damage to brain neurons [36–39]. Ceramide activates Toll-like receptor 4, which leads to nuclear factor kappa B activation and inflammatory responses [40]. Toll-like receptor 4 is a transmembrane protein that activates signaling pathways through myeloid differentiation primary response gene 88 and TIR domain-containing adaptor protein. These complex signaling pathways lead to inflammatory responses, including inflammatory cytokine production.

Ceramide inhibits vascular endothelial cell-induced angiogenesis, which compromises repair of the blood–brain barrier [41]. Transport pumps in the blood–brain barrier are activated by ceramide, which may increase damage to the brain [34,35] by enhancing the uptake of damaging molecules into the brain.

Homocysteine induces the synthesis of ceramide in the brain and enhances ceramide-induced oxidative stress in the brain [42]. This could enhance ceramide-induced damage to the blood–brain barrier and the brain. Hyperhomocysteinemia is discussed later in this manuscript.

## 4. Endocannabinoids

Many people use marijuana, *Cannabis sativa*, to self-treat their dementia and Alzheimer's disease. Several studies have not found evidence for the efficacy of *C. sativa* for this use [43,44]. A population-based study of late-onset Alzheimer's disease found that *C. sativa* decreases dementia-related symptoms [45]. To date, there is no question that *C. sativa* is safe to use in Alzheimer's disease.

There are no changes in the endocannabinoids, anandamide or 2-arachidonoyl glycerol brain levels in the brains of Alzheimer's disease patients compared to controls [46]. Cannabinoid receptor 1 (CB1) expression does not appear to change due to Alzheimer's disease [46]. Fatty acid amide hydrolase deactivates anandamide and is enhanced in the brains of Alzheimer's disease patients [46]. This suggests that the synthesis of anandamide may be high in Alzheimer's disease but is kept in the normal range by enhanced fatty acid amide hydrolase. CB2 expression increases in the microglial cells of brains from Alzheimer's disease patients [46]. CB2 expression and microglial activation are involved in the neuroinflammation that causes Alzheimer's disease. This suggests that the use of a CB2 inhibitor, cannabidiol, may be useful to slow down the progression of Alzheimer's disease. The low oral bioavailability of cannabidiol may limit its usefulness. To date, there is no convincing evidence that endocannabinoids are involved in causing Alzheimer's disease.

## 5. Adipokines

Visceral fat secretes inflammatory adipokines into the blood that cause type 2 diabetes, heart disease, arthritis and other problems [36]. Visfatin is a visceral fat-derived inflammatory adipokine (Table 2), as is monocyte chemoattractant protein-1. Visfatin works with xanthine oxidase and NADH oxidase to increase oxygen radical production in the lumens of capillaries, which damages the blood–brain barrier [36–39]. Since visfatin depletes blood nicotinamide, this may diminish the uptake of nicotinamide into the brain and deplete brain NAD. Monocyte chemoattractant protein-1 allows monocytes to adhere to damaged endothelial cells and cross the blood–brain barrier to enhance brain inflammation.

Several inflammatory adipokines are elevated in the blood of patients with Alzheimer's disease [2], including dipeptidyl peptidase-4, IL-1β, IL-2, IL-6, IL-18, interferon-γ, C-reactive protein, CXC chemokine-10, epidermal growth factor, vascular cell adhesion protein1, tumor necrosis factor α and leptin. Dipeptidyl peptidase-4 enhances vascular aging and may increase blood–brain barrier damage [47]. IL-1β and IL-6 induce hepcidin transcription [48], which causes iron overload in neurons during neuroinflammation, possibly increasing neuronal death [49]. C-reactive protein elevation is predictive of death in frail patients [50]. C-reactive protein binds to lysophosphatidylcholine on damaged cells, including endothelial cells, and activates the complement system. It also induces NADPH oxidase on the luminal membranes of endothelial cells of the blood–brain barrier, which enhances oxygen radical formation and damages the blood–brain barrier [51]. Chemokines and vascular cell adhesion protein1 increase the adhesion and brain penetration of monocytes and neutrophils. Homocysteine is higher in the blood of βpatients with Alzheimer's disease than controls [2]. Nitric oxide synthase dysfunction and oxygen radical production occur due to high blood homocysteine with consequent damage to the microvasculature [52]. This damages the blood–brain barrier, which allows essential compounds to leak out of the brain and harmful compounds and inflammatory cells to enter into the brain.

High blood IL-1β levels are involved in causing Alzheimer's disease. IL-1β increases blood–brain barrier permeability by inducing hypoxia-inducible factor-1α and vascular endothelial growth factor-A [53]. This allows toxic substances to enter the brain and essential nutrients to exit the brain. This leakiness may be involved in the induction of Alzheimer's disease and other neurodegenerative disorders. High blood tumor necrosis factor α levels are involved in causing Alzheimer's disease. Tumor necrosis factor α crosses the blood–brain barrier through a receptor-mediated process [54]. It also increases the permeability of other molecules across the blood–brain barrier.

**Table 2.** Adipokines, homocysteine and ceramide are involved in Alzheimer's disease.

| Factor | Receptor/Substrate | Effect |
|---|---|---|
| Ceramide | Ceramide activated serine/threonine phosphatases | Toll-like receptor 4, endothelial nitric oxide synthase |
| Homocysteine | N-methyl-D-aspartate receptor on endothelial cells | Endothelial nitric oxide synthase |
| Visfatin | Nicotinamide | Increases extracellular NADH levels, which makes oxygen radicals due to NADH oxidase |
| Dipeptidyl peptidase-4 | Cleaves proline or alanine dipeptides from many proteins | Endothelial damage |
| IL-1β | IL-1β receptor | Blood–brain barrier damage |
| C-reactive protein | Lysophosphatidylcholine | Induces NADPH oxidase damages blood–brain barrier |
| Tumor necrosis factor α | TNF receptor | Blood–brain barrier damage |

## 6. Diet

Dietary changes in the aging have been noted in several studies [55–58]. These dietary changes probably explain why up to 40% of elderly people do not eat diets that contain adequate amounts of nicotinamide (Table 3), which is a form of vitamin B3 [59]. Nicotinamide is neuroprotective [60]. Nicotinic acid intake has been shown to be inversely related to the incidence of Alzheimer's disease [61]. Nicotinic acid is converted into nicotinamide by nicotinamidase before it can be taken up into the brain [60]. However, increased blood high-density lipoprotein by nicotinic acid may be beneficial to the blood–brain barrier. The metabolite of nicotinamide, N-methylnicotinamide, protects the vasculature and decreases homocysteine levels [62]. Inadequate dietary intake of magnesium occurs in 60% of people due to inadequate intake of fruits and vegetables [63]. Chocolate is a good source of magnesium, with 41 mg of magnesium per 28 g of dark chocolate. Magnesium protects the vasculature against atherosclerotic changes [64].

N-3 polyunsaturated fat consumption, α linolenic acid C18:3N-3, eicosapentaenoic acid C20:5N-3, docosahexaenoic acid C22:6N-3, may benefit the brain but does not decrease the incidence or progression of Alzheimer's disease [65]. A-Linolenic acid is from flaxseed, chia seeds, soybean and canola oil. Eicosapentaenoic acid and docosahexaenoic acid are from fish and marine microalgae. Adequate daily intake of α linolenic acid in adults above 51 years old is 1.6 g per day for men and 1.1 g per day for women.

Inadequate dietary folate is involved in causing Alzheimer's disease. Low intake of folate, vitamin B9, increases the risk of developing Alzheimer's disease and the severity of atrophy of the cerebral cortex [66]. This may relate to the ability of folate to decrease blood homocysteine levels [67]. Folate protects endothelial cells and may benefit the blood–brain barrier [68]. Folate is abundant in leafy green vegetables. The daily recommended intake of folate is 400 μg. Spinach, 30 g, contains 58.2 μg of folate. Fruit can also contain folate. One orange may contain 55 μg of folate. Vegetables are also good sources of folate, including asparagus, Brussels sprouts and broccoli. Blueberries are rich in folate, 24 μg per 340 g, and nicotinic acid 1.7 mg per 340 g. Strawberries are also good folate sources, 100 μg per 100 g of berries, and contain some nicotinic acid 0.5 mg per 100 g of berries. Recent studies have found diets that contain adequate leafy green vegetables and fruits slow age-related cognitive decline and may decrease the onset of Alzheimer's disease [69,70]. Of course, folate, as well as flavonoids, betulinic acid and other natural compounds, are beneficial in these diets [37,71].

**Table 3.** Dietary factors involved in Alzheimer's disease.

| Depleted Nutrient | Dietary Source | Effects |
|---|---|---|
| Magnesium | Chocolate, vegetables, nuts | Endothelial damage |
| Nicotinamide | Vegetables, meat, milk, eggs | Brain atrophy, vascular damage |
| Folate | Vegetables, fruits | Brain atrophy |

## 7. Kynurenine/NAD

The adipokines involved in causing Alzheimer's disease to alter the activities of enzymes in the kynurenine pathway. Tryptophan is converted into NAD by a series of reactions called the kynurenine pathway [72–75]. Tryptophan is oxidized to make kynurenine through the intermediate N-formyl kynurenine. 3-Hydroxykynurenine is the next metabolite and is oxidized to make 3-hydroxanthranilic acid, then 2-amino-3-carboxymuconate semialdehyde, then quinolinic acid, which makes nicotinic acid. Intracellular nicotinic acid is quickly and completely converted to nicotinamide, which is the main precursor for NAD [38,60]. Cellular energy depends on ATP, NAD, NADH, NADPH, cAMP and other factors. NAD is the energy switch that responds to energy deficits or excesses and adjusts protein activities and genetic transcription appropriately. Energy is involved in DNA repair, mitochondrial electron transport, energy regulation and other processes. NAD switches cellular energy processes through two enzymes, poly (ADP-ribose) polymerase and sirtuins. Sirtuins use NAD to deacetylate proteins and alter their activities [76]. Poly (ADP-ribose) polymerase uses NAD to poly (ADP-ribosylate) other proteins and alters their activities [77,78]. Poly (ADP-ribose) polymerase is one of the most important proteins involved in DNA repair. Damage to DNA can cause the depletion of NAD and ATP [77,78].

NAD is a high-energy compound that contains high-energy phosphate bonds and adenine. ATP has similar characteristics and is synthesized from the same precursors as NAD. Depletion of NAD by excessive activity of sirtuins or poly (ADP-ribose) polymerase also depletes ATP [79–81]. This depletion of cellular energy makes cells more vulnerable to oxidative stress, other challenges and can cause cell death.

Blood tryptophan levels can be low in Alzheimer's disease patients [82,83], which may lead to low brain NAD levels. Brain NAD depletion is implicated in several animal models of Alzheimer's disease [84]. About 40% of elderly people do not eat enough nicotinamide or niacin, which are NAD precursors [59]. People who eat inadequate amounts of niacin and nicotinamide are prone to developing Alzheimer's disease [61]. Depletion of brain NAD has a number of consequences: impaired neurogenesis, altered telomere shortening, increased neuroinflammation, decreased DNA repair, impaired lysosomal activity, impaired proteasomal activity and mitochondrial dysfunction [84].

Alzheimer's disease patients have elevated activities of several adipokines that alter kynurenine pathway enzyme activities, potentially decreasing NAD synthesis. IL-1β and tumor necrosis factor α are elevated in Alzheimer's disease and induce the activity of kynurenine 3-monooxygenase [72–75]. This enzyme makes 3-hydroxykynurenine from kynurenine. Chronic activation of this enzyme results in reactive oxygen species production, including hydrogen peroxide [72–75]. This oxidative stress damages the brain. Increased blood 3-hydroxykynurenine levels have been reported in Alzheimer's disease [85]. Indoleamine 2,3-oxygenase is induced by IL-1β and tumor necrosis factor α [75]. This enzyme makes N-formyl kynurenine from tryptophan. NAD is made from quinolinic acid, a metabolite of tryptophan. It is not known if NAD synthesis is normal in Alzheimer's disease brains.

Several tryptophan metabolites are neurotoxic and may be involved in causing Alzheimer's disease [72–75]. 3-Hydroxy-L-kynurenine, 3-hydroxyanthranilic acid and quinolinic acid are neurotoxic. Quinolinic acid is an agonist for N-methyl-D-aspartate (NMDA) receptors and is excitotoxic [72–75]. 3-Hydroxy-L-kynurenine and 3-hydroxyanthranilic acid stimulate reactive oxygen radical formation [72–75]. Excessive production of these

toxic metabolites and reactive oxygen species may damage neurons, glial cells and the blood–brain barrier.

## 8. MicroRNA

Over 2000 microRNAs have been identified and are known to have extensive activities in post-translational gene control [86]. MicroRNAs are clearly involved in the neurodegeneration that leads to Alzheimer's disease. Several microRNAs are downregulated in Alzheimer's disease: MiR-16, MiR-17, MiR-33, MiR-206, MiR-330 [86]. Other microRNAs are upregulated in Alzheimer's disease: MiR-26b, MiR-34a, MiR-98 [86]. These microRNAs are involved in nutrient sensing, senescence, the regulation of NADPH oxidase, sirtuin 4, cyclooxygenase, Toll-like receptor, c-Jun N-terminal kinase, extracellular signaling kinase 1 and the regulation of many other enzymes and genes [86].

Ceramide synthesis is regulated by MiR-34a, which inhibits the activity of ceramide kinase, resulting in ceramide accumulation [87]. Ceramide then has several activities in causing Alzheimer's disease, as previously discussed.

Folate is involved in the regulation of DNA repair, synthesis and methylation [88]. Folate also regulates microRNA synthesis [87], upregulating some microRNAs and downregulating other microRNAs. MicroRNAs then regulate the activities of enzymes involved in folate metabolites [88].

Tumor necrosis factor $\alpha$ upregulates microRNA synthesis. For instance, MiR-155 synthesis is upregulated, which enhances the activity of sirtuin 1 and causes the senescence of endothelial cells, such as in the blood–brain barrier [89]. On the other hand, microRNA can upregulate the transcription of tumor necrosis factor $\alpha$ [90].

## 9. Depression, Sleep and Alzheimer's Disease

It has been clear for many years that there may be a relationship between depression Alzheimer's disease [91,92]. It has proved impossible so far to show that depression causes Alzheimer's disease [91–93]. Antidepressant therapy has not been shown to be useful in the treatment of Alzheimer's disease [94]. Microglia secrete inflammatory cytokines that cause brain inflammation that may be associated with Alzheimer's disease and major depression [95]. It is not clear if microglial activation causes Alzheimer's disease or is a consequence of damage to the blood–brain barrier, as discussed above.

Sleep disturbances are common in Alzheimer's disease, in elderly people and in depression [96–99]. There is no convincing evidence that sleep disturbances cause Alzheimer's disease. However, the inflammatory adipokines, IL1 and tumor necrosis factor $\alpha$, that are involved in causing Alzheimer's disease also cause sleep disturbances [100]. The two conditions share common mechanisms of induction.

## 10. Discussion

It is probable that Alzheimer's disease is caused by sedentary lifestyles, visceral fat accumulation, inadequate exercise and poor nutrition, especially in terms of fruit and vegetable intake. All of this leads to damage to the blood–brain barrier, neuronal death and inflammation of the brain that are hallmarks of Alzheimer's disease.

Prevention of Alzheimer's disease involves daily exercise, maintaining a lean body with little visceral fat, and eating fruits and vegetables daily. Healthcare professionals can do a great service to their patients by teaching them how to prevent Alzheimer's disease: exercise daily, maintain a lean body, eat green leafy vegetables, other vegetables and fruit.

As people grow older, age-related sarcopenia occurs [13]. This makes many people stop exercising and leads to visceral fat accumulation. The lack of exercise depletes the body of lactic acid that is essential to brain nutrition, and is a signaling molecule [15]. Inadequate lactic acid results in damage to neurons and the blood–brain barrier. Visceral fat secretes ceramide and adipokines into the blood, such as IL-1$\beta$ and tumor necrosis factor $\alpha$ [2]. Homocysteine levels are also high in the blood of Alzheimer's disease patients [2]. These factors damage the blood–brain barrier and the brain. Age-related changes in dietary

preferences lead to inadequate folate intake in many elderly people [66]. Folate deficiency leads to damage to the blood–brain barrier [90]. Inadequate intake of nicotinamide may deplete brain NAD leading to changes in the kynurenine pathway and damage to the brain [71–74].

MicroRNA synthesis changes due to a cascade of events involving adipokines, ceramide and folate [86–89]. Changes in microRNA alter brain biochemistry leading to brain damage. Alzheimer's disease is caused by this cascade of events. Many of these events can be prevented, which makes Alzheimer's disease preventable.

**Funding:** This research received no external funding.

**Institutional Review Board Statement:** Not applicable.

**Informed Consent Statement:** Not applicable.

**Data Availability Statement:** Not applicable.

**Conflicts of Interest:** The author declares no conflict of interest.

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
