# Peer review of "Probable Causes of Alzheimer’s Disease"

_sci, doi:10.3390/sci3010016_

Round 1

Reviewer 1 Report

In this review, the author discussed the possible causes of Alzheimer’s disease (AD) in terms of everyday diet, lifestyle, where interesting molecular markers of AD were also correspondingly discussed such as lactic acid, ceramide, endocannabinoids, adipokines. The present paper here indicated that exercises and diets were also attributed to the onset of AD apart from the most well-studied biomolecules like amyloid-beta proteins, tau proteins and phosphorylated tau. Overall, this is an interesting paper focusing on discussing causes of AD and their relations. However, there are several concerns regarding the discussion and literature cited in this paper. I can recommend publication after the respective concerns were addressed.

Open questions:

  • Are there any straightforward evidence demonstrating health diet would lowering the risk of developing AD excluding genetic causes?
  • In the aging society, are the diet and exercising habits good category to divide individuals since there would be little difference? Are molecular markers discussed here truly related to diet and exercising appear more likely in the younger community?

Major recommendations:

  1. The author presented the relations between the risky factors and AD but the lack of molecular evidence or published literature to demonstrate that they were the potential causes of AD. Therefore, it had better alter “causes” to “factors”.
  2. In Abstract, the author presented the three-part mechanism in inducing AD and its significance in indication of individual difference in developing AD. However, the author omitted some previous reviews on the how diet and exercise reduce AD risk and did not presented the novelty and focus of this review.
  3. In Introduction, the author pointed out the failure of current academic research and clinical trials on Amyloid beta and Tau protein, two potential misfolded proteins leading to AD but simply claimed that the wrong direction was chosen for treating and preventing AD. However, this conclusion is lacking solid scientific evidence and sufficient literature citations. Therefore, more discussions should be provided regarding the claim. Also, the rationale of focusing on diet and exercises was expected to make the introduction more convincing and scientifically sound.
  4. The author provided the solid evidence in supporting the essential role of lactic acid in muscle-brain axis. However, the claim that “There is no question that frailty increases the risk of developing Alzheimer’s disease” is not fully supported by the reference 11 which examined the association between physical frailty and beta amyloid deposits. From this point of view, the reference is also contradictory to the one conclusion in the introduction that “the amyloid-tau theories of Alzheimer’s disease induction are wrong”. It would be of great help if the author could revise this part further.
  5. The factor ceramides presented by author is associated with AD but there is no strong data to show that it is also a potential cause of AD. Therefore, either the title should be altered or the subheading of ceramides need to be further revision.
  6. The second paragraph in the subheading of “Endocannabinoids” only used one reference in total. Are there any other appropriate references the author could provide here to validate the importance of endocannabinoids in AD?
  7. The author presented that inflammatory agents adipokines contributed greatly to blood brain barriers in detail, however, its association with AD is lacking evidence, references and discussion. It is recommended adding such comments considering the readership of the journal.

Minor recommendations:

  1. Subheadings should have the same font size. The second subtitle looks bigger than the others somehow. It might be due to the formatting of the manuscript.
  2. Sentences seem independent but actually related. Therefore Line 65-68 needs more conjunction words.
  3. A scheme of how these factors related to each other and AD would be better in presenting the different relations and factors, as well as great for the comprehensive readership.

Author Response

  1. The dietary studies by McGrattan and Morris show that dietary changes lower the risk of developing AD as stated in the manuscript.
  2. I have included a new reference about physical activity and AD by Stephen.
  3. It is very easy to distinguish between diet in the elderly and young.  I have included new sections on kynurenine/NAD to help with this.
  4. I have presented more than enough molecular evidence to show the probable causes of AD, including a new section on microRNA.
  5. I have included reviews that discuss all of the previous studies on diet and physical activity.
  6. I have added references on the failure of amyloid beta therapies.  The rational for diet and exercise is obvious from the fact that AD can be prevented with these changes.
  7. I have changed this reference as suggested to the Stephen study.
  8. I have expanded the ceramide section.
  9. There is no good evidence to show that endocannabinoids cause AD.  The purpose of this section is to dissuade patients from using marijuana to self treat AD.
  10. I include a table on adipokines in AD.
  11. I have included 3 new tables to help with the clarity of the work.

Reviewer 2 Report

2 February 2021

Review on the manuscript titled “Probable Causes of Alzheimer’s Disease” by Adams J, submitted to Sci.

Dear Author,

The author proposed a three-part mechanism of the pathogenesis of Alzheimer’s disease (AD): decreased blood lactic acid, increased blood ceramides, and decreased blood folic acid, concluding that simple changes in lifestyle and diet can prevent Alzheimer’s disease.

Please reconsider the following parts:

  1. Page 1, Abstract: Please expand the abstract to 200 words.
  2. Page 1, Introduction: In addition to the amyloid-tau hypothesis, there are other hypotheses for the pathogenesis of AD. Please introduce them and add more descriptions on risk factors and comorbidity of AD leading to main topics of the manuscript. Suggested references: Tanaka, M.; Bohár, Z.; Vécsei, L. Are Kynurenines Accomplices or Principal Villains in Dementia? Maintenance of Kynurenine Metabolism. Molecules202025, 564. Tanaka, M.; Toldi, J.; Vécsei, L. Exploring the Etiological Links behind Neurodegenerative Diseases: Inflammatory Cytokines and Bioactive Kynurenines.  J. Mol. Sci. 202021, 2431.
  3. Pages 1, 2, Exercise/Lactic Acid: Please plot a figure or table to summarize the section.
  4. Page 2, Ceramide: This section is too short. Please add more information on ceramides including physiology in healthy individuals, AD patients and drug delivery through the blood-brain barrier. And summarize in a table or figure. Suggested reference: Di Biase, E.; Lunghi, G.; Maggioni, M.; Fazzari, M.; Pomè, D.Y.; Loberto, N.; Ciampa, M.G.; Fato, P.; Mauri, L.; Sevin, E.; Gosselet, F.; Sonnino, S.; Chiricozzi, E. GM1 Oligosaccharide Crosses the Human Blood–Brain Barrier In Vitro by a Paracellular Route.  J. Mol. Sci.202021, 2858.
  5. Page 3, Endocannabinoids: Please add more information on endocannabinoids associated with AD.
  6. Page 3, Adipokines: Please add a figure or table summarizing the section.
  7. Pages 3-4, Diet: Please add a figure or table summarizing the section.
  8. Page 4, Discussion: Please expand the discussion including topics described in the previous sections to develop arguments.
  9. References: Please cite more references. Here is a list of reference which deserves to cite:

Török, N.; Tanaka, M.; Vécsei, L. Searching for Peripheral Biomarkers in Neurodegenerative Diseases: The Tryptophan-Kynurenine Metabolic Pathway. Int. J. Mol. Sci. 2020, 21, 9338.

Verma, V.; Singh, D.; KH, R. Sinapic Acid Alleviates Oxidative Stress and Neuro-Inflammatory Changes in Sporadic Model of Alzheimer’s Disease in Rats. Brain Sci. 202010, 923.

O’Day, D.H. Calmodulin Binding Proteins and Alzheimer’s Disease: Biomarkers, Regulatory Enzymes and Receptors That Are Regulated by Calmodulin. Int. J. Mol. Sci. 202021, 7344.

Catanesi, M.; d’Angelo, M.; Tupone, M.G.; Benedetti, E.; Giordano, A.; Castelli, V.; Cimini, A. MicroRNAs Dysregulation and Mitochondrial Dysfunction in Neurodegenerative Diseases. Int. J. Mol. Sci. 202021, 5986.

Brito, L.M.; Ribeiro-dos-Santos, Â.; Vidal, A.F.; de Araújo, G.S., on behalf of the Alzheimer’s Disease Neuroimaging Initiative; Differential Expression and miRNA–Gene Interactions in Early and Late Mild Cognitive Impairment. Biology 20209, 251.

Cantón-Habas, V.; Rich-Ruiz, M.; Romero-Saldaña, M.; Carrera-González, M.P. Depression as a Risk Factor for Dementia and Alzheimer’s Disease. Biomedicines 20208, 457.

The manuscript contains no figure, no table, and 61 references. The reviewer recommends including more figures and tables to convey the most important parts of topics. The word count should be at least 5000 and the number of references should be more than 150. The manuscript is inconsistent with contents described in Abstract and Main Body. For example, folic acid is not discussed at all. And main concept presented in Abstract such as three-part mechanism is not supported and discussed in the manuscript. The manuscript carries potentially valuable information. Therefore, I reconsider this manuscript for publication after major revision.

Best regards,

Author Response

I have extensively altered the manuscript as shown in yellow.  I have included almost all of the references you suggest.  

  1. The abstract is expanded.
  2. I have included a new section on kynurenine/NAD.
  3. I have added 3 new tables.
  4. I have expanded the ceramide section.
  5. I cannot find good evidence that endocannabinoids cause or prevent AD.  I included this section to dissuade patients from self treating with marijuana.
  6. I have added tables for adipokines and diet.
  7. I have expanded the discussion.
  8. I have added many more references.

Round 2

Reviewer 1 Report

The author has answered the all of the concerns and considerations in the first report. It is good for publication. 

Author Response

I appreciate your help making it a better manuscript.

Reviewer 2 Report

15 February 2021

The second review on the manuscript titled “Probable Causes of Alzheimer’s Disease” by Adams J, submitted to Sci.

Dear Author,

The author proposed a three-part mechanism of the pathogenesis of Alzheimer’s disease (AD): decreased blood lactic acid, increased blood ceramides, and decreased blood folic acid, concluding that simple changes in lifestyle and diet can prevent Alzheimer’s disease.

Please reconsider the following parts:

  1. Page 1, Abstract: Please summarize in the end of Abstract what readers are going to read in the article.
  2. The Body: Please discuss three-part mechanism presented in Abstract in the body of the manuscript.
  3. References: Please cite more references to make at least more than 100. Here is a suggested list of references:

Muntsant, A.; Jiménez-Altayó, F.; Puertas-Umbert, L.; Jiménez-Xarrie, E.; Vila, E.; Giménez-Llort, L. Sex-Dependent End-of-Life Mental and Vascular Scenarios for Compensatory Mechanisms in Mice with Normal and AD-Neurodegenerative Aging. Biomedicines 2021, 9, 111. https://doi.org/10.3390/biomedicines9020111

Kim, J.; Kim, Y.-K. Crosstalk between Depression and Dementia with Resting-State fMRI Studies and Its Relationship with Cognitive Functioning. Biomedicines 20219, 82. https://doi.org/10.3390/biomedicines9010082

O’Day, D.H. Calmodulin Binding Proteins and Alzheimer’s Disease: Biomarkers, Regulatory Enzymes and Receptors That Are Regulated by Calmodulin. Int. J. Mol. Sci. 202021, 7344.

Brito, L.M.; Ribeiro-dos-Santos, Â.; Vidal, A.F.; de Araújo, G.S., on behalf of the Alzheimer’s Disease Neuroimaging Initiative; Differential Expression and miRNA–Gene Interactions in Early and Late Mild Cognitive Impairment. Biology 20209, 251.

Cantón-Habas, V.; Rich-Ruiz, M.; Romero-Saldaña, M.; Carrera-González, M.P. Depression as a Risk Factor for Dementia and Alzheimer’s Disease. Biomedicines 20208, 457.

The manuscript contains no figure, three table, and 90 references. The manuscript is substantially improved. The word count is 3800 which is less than 5000. The manuscript carries valuable information. Therefore, I accept this manuscript for publication after minor revision.

Best regards,

Author Response

  1. I have made sure the abstract summarizes what the readers will find in the manuscript.
  2. I have made sure the 3 part mechanism is discussed in the body of the manuscript.
  3. I have written a section on depression and sleep.  There are now 100 references.

Round 3

Reviewer 2 Report

17 February 2021

The third review on the manuscript titled “Probable Causes of Alzheimer’s Disease” by Adams J, submitted to Sci.

Dear Author,

The author proposed a three-part mechanism of the pathogenesis of Alzheimer’s disease (AD): decreased blood lactic acid, increased blood ceramides, and decreased blood folic acid, concluding that simple changes in lifestyle and diet can prevent Alzheimer’s disease.

The manuscript contains no figure, three table, and 100 references. The section of depression and sleep was added. The manuscript is substantially improved. The manuscript carries valuable information. Therefore, I accept this manuscript for publication in present form.

Best regards,